# Detection of *Mycobacterium tuberculosis* from tongue swabs using sonication and sequence-specific hybridization capture

**Alexander J. Yan**[1]*, **Alaina M. Olson**[2], **Kris M. Weigel**[2], **Angelique K. Luabeya**[3], **Erin Heiniger**[1], **Mark Hatherill**[3], **Gerard A. Cangelosi**[2], **Paul Yager**[1]

1 Department of Bioengineering, University of Washington, Seattle, Washington, United States of America, 2 Department of Environmental and Occupational Health Sciences, University of Washington, Seattle, Washington, United States of America, 3 South African Tuberculosis Vaccine Initiative, Institute of Infectious Disease & Molecular Medicine and Department of Pathology, Faculty of Health Sciences, University of Cape Town, Cape Town, South Africa

* ajyan@uw.edu

**Data Availability Statement:** All relevant data are within the paper and its Supporting information files.

**Funding:** Study funding was provided by the Bill & Melinda Gates Foundation (INV-004527) and the

## Abstract

Tongue swabs hold promise as a non-invasive sample for diagnosing tuberculosis (TB). However, their utility as replacements for sputum has been limited by their varied diagnostic performance in PCR assays compared to sputum. The use of silica-based DNA extraction methods may limit sensitivity due to incomplete lysis of *Mycobacterium tuberculosis* (MTB) cells and co-extraction of non-target nucleic acid, which may inhibit PCR. Specificity may also be compromised because these methods are labor-intensive and prone to cross-contamination. To address these limitations, we developed a sample preparation method that combines sonication for MTB lysis and a sequence-specific MTB DNA capture method using hybridization probes immobilized on magnetic beads. In spiked tongue swabs, our hybridization capture method demonstrated a 100-fold increase in MTB DNA yield over silica-based Qiagen DNA extraction and ethanol precipitation. In a study conducted on clinical samples from South Africa, our protocol had 74% (70/94) sensitivity and 98% (41/42) specificity for detecting active pulmonary TB with sputum Xpert MTB/RIF Ultra as the reference standard. While hybridization capture did not show improved sensitivity over Qiagen DNA extraction and ethanol precipitation, it demonstrated better specificity than previously reported methods and was easier to perform. With integration into point-of-care platforms, these strategies have the potential to help enable rapid non-sputum-based TB diagnosis across key underserved patient populations.

## Introduction

Tuberculosis (TB) remains a significant global health issue and is a leading infectious cause of global mortality despite the availability of effective treatments. In 2022, there were an estimated 10.6 million TB infections and 1.3 million deaths globally, highlighting the need for rapid and accurate diagnostic tools to mitigate the spread of the causative agent *Mycobacterium*

National Institutes of Health (U54EB027049 and R01AI139254) to JC. AY received funding from the National Science Foundation Graduate Research Fellowship Program. The funders had no role in study design, data collection and analysis, decision to publish, or preparation of the manuscript.

**Competing interests:** I have read the journal's policy and the authors of this manuscript have the following competing interests: Paul Yager has a nonpaying appointment as CSO of the UbiDX corporation. There are no patents, products in development, or marketed products associated with this research to declare. Swabs were kindly donated by Copan. This does not alter our adherence to PLOS ONE policies on sharing data and materials.

*tuberculosis* (MTB) [1]. Sputum is the standard patient sample for most TB diagnostic tests, which include smear microscopy, microbiological culture, and nucleic acid amplification tests such as the Cepheid Xpert® MTB/RIF Ultra (Xpert Ultra). However, sputum is challenging to produce for certain patient groups, including children and those living with HIV, leaving these populations vulnerable to underdiagnosis [2]. Moreover, sputum collection generates potentially infectious aerosols, posing a hazard to healthcare workers and others present at the collection site. Given these limitations, the development of a non-sputum-based TB test is considered a critical step toward broadening access to TB diagnosis and improving TB screening in community settings [3].

Recent studies by our consortium and others have demonstrated that MTB cells and/or their DNA accumulate on the oral epithelium during active TB and can be collected for analysis with oral swabs [4–10]. Oral swabbing (OS) is performed by brushing a disposable swab against the dorsum of the tongue and eluting collected material into sample buffer for nucleic acid amplification targeting MTB DNA. OS is easy to perform, poses a minimal occupational health hazard, and is preferred over sputum collection by health care workers [11]. Importantly, as sputum production is not required, OS enables diagnosis for sputum-scarce patients, and facilitates active case finding to screen for TB outside of traditional healthcare facilities.

While OS has been considered a potential alternative sampling method for diagnosing TB, recent studies on OS with manual DNA extraction methods have produced varying results. In a blinded study conducted on 219 adults with TB in South Africa, OS demonstrated sensitivity relative to sputum Xpert Ultra of 88% and 94%, respectively, from swabs collected on two separate days [7]. However, specificity was low at 79%. In a more recent study conducted on 64 South African people living with HIV and TB, sensitivity from a single swab was lower at 67%, and specificity remained subpar at 78% [9]. In these studies, specificity may have been adversely affected by the use of manual, in-house DNA extraction methods. Specificities have been better when automated methods such as the Xpert Ultra were used to test swabs. For example, Andama et al. [10] reported 100% specificity and 78% sensitivity for OS Xpert Ultra, relative to sputum Xpert Ultra.

Clinical accuracy limitations reported in previous OS studies may be due in part to sample preparation protocols that are suboptimal for tongue swabs. Oral samples contain a high concentration of non-target bacterial nucleic acids that may inhibit the amplification and detection of target MTB DNA. Conventional silica-based DNA extraction methods (e.g., Qiagen DNA spin column kits) used in many past studies [4–9] rely on the adsorption of DNA to silica under chaotropic conditions, resulting in the inadvertent co-extraction of non-target DNA. Furthermore, MTB DNA extraction efficiency may be limited by lysis methods that are inefficient for MTB, which is notably resistant to conventional chemical (e.g., guanidinium salt) and enzymatic (e.g., proteinase K) methods of lysis [12]. Finally, ethanol precipitation and wash protocols used in past studies [4–9] may have enabled laboratory cross-contamination of samples, thereby reducing diagnostic specificity.

Several strategies have been developed to reduce the co-extraction of non-target nucleic acid. A commonly used approach uses oligonucleotide probes that are complementary to specific target sequences and bind to their targets through hybridization. These probes are bound to a solid substrate, typically magnetic beads, to facilitate isolation of target DNA and removal of background nucleic acid. This strategy concentrates target DNA into a smaller volume, thereby enhancing assay sensitivity. By combining DNA extraction with DNA concentration, this strategy eliminates the need for a separate DNA concentration process (i.e., ethanol precipitation). This streamlined approach simplifies sample processing for the operator,

minimizes the risk of cross-contamination of samples, and has the potential to improve overall accuracy.

In this paper, we report a method to selectively purify MTB DNA from tongue swab samples using a mechanical lysis step combined with MTB DNA sequence-specific capture (Fig 1). Tongue swab samples are first heated to deactivate pathogens and then sonicated to fully lyse MTB cells. MTB-specific biotinylated oligonucleotides are then added to the lysed oral sample for capture via hybridization, followed by incubation with streptavidin-coated magnetic beads to isolate and concentrate MTB DNA. While hybridization capture has previously been utilized for TB detection from a variety of sample matrices including sputum, urine, and bronchoalveolar lavage fluid [13–15], this study is the first to demonstrate its use with TB tongue swab samples. Here, we sought to accomplish two goals: 1) optimize the hybridization capture method for use with tongue swabs; and 2) determine the diagnostic accuracy of our method in clinical tongue swab specimens from adults with active pulmonary TB.

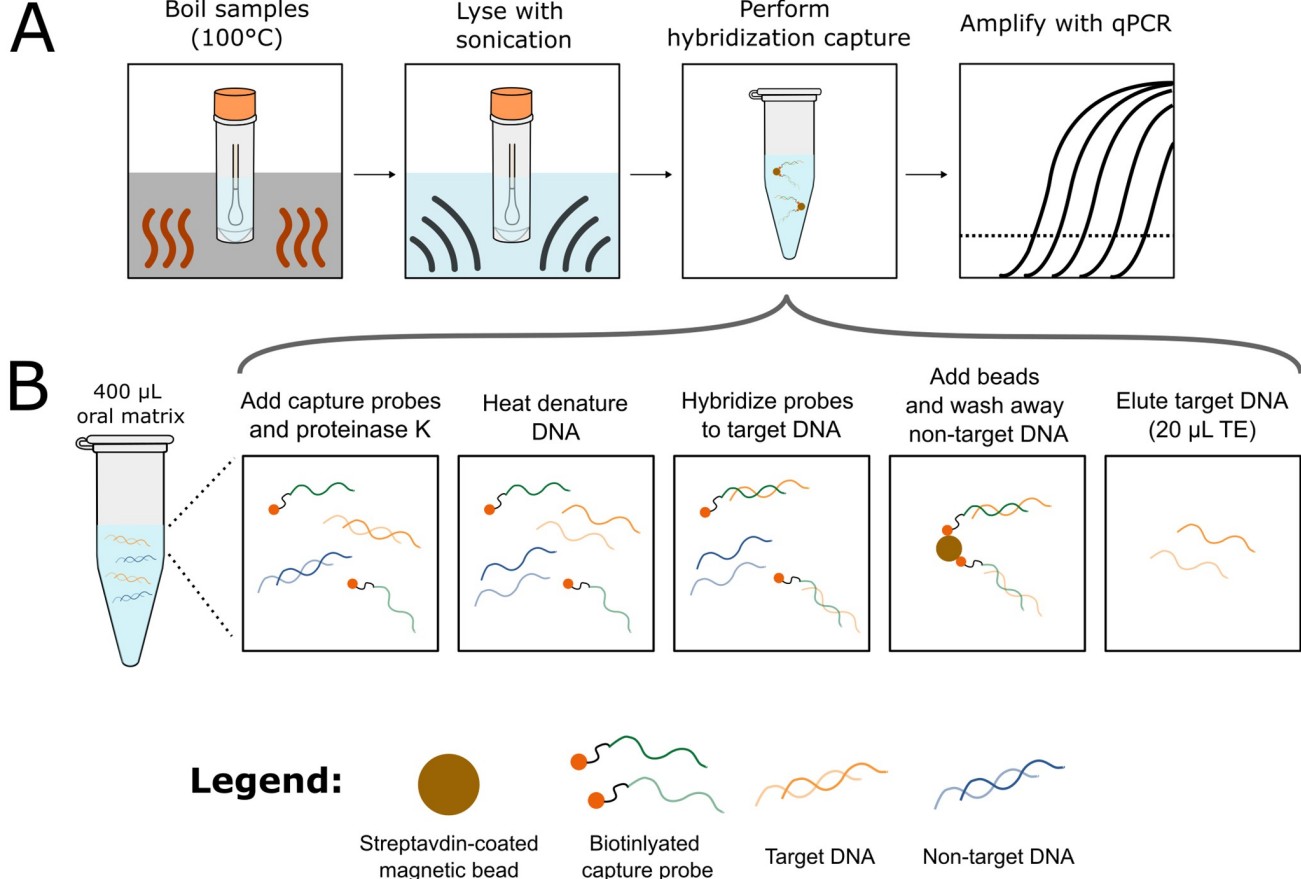

**Fig 1. Workflow of MTB DNA extraction from tongue swabs.** (**A**) Schematic overview of the MTB lysis and DNA extraction protocol. Tongue swabs in lysis buffer are first heated (10 min at 100˚C) to kill pathogens and inactivate nucleases. Once cooled, samples are lysed in a sonication bath for 10 min, and then subjected to the hybridization capture procedure. Finally, the purified DNA samples are amplified by qPCR. (**B**) Overview of the hybridization capture procedure. Biotinylated capture probes and proteinase K are incubated with lysed oral samples (30 min at 55˚C), heat denatured (10 min at 100˚C), and hybridized to target MTB DNA (20 min at 55˚C). Streptavidin-coated magnetic beads are added to capture target DNA (1 hr at RT) and enable washing of non-target DNA and other PCR inhibitors. Finally, target DNA is eluted from the capture beads using heat (5 min at 95˚C). Panel B is adapted from Oreskovic and Lutz [16].

## Materials and methods

### Methods for DNA extraction in spiked tongue swabs

**Culture of MTB cells.** H37Ra MTB stocks (ATCC, Manassas, VA, USA) were inoculated in 50-mL conical tubes containing 10 mL growth medium (Middlebrook 7H9, 0.2% glycerol, 40 mM sodium pyruvate, 10% ADC, 0.05% Tween-80) and rotated in a drum at 37°C, with lids slightly cracked. Cell concentrations were determined by measuring optical density (OD600) 7–10 days post-inoculation. Cells were diluted down to desired concentrations using PBSGT buffer (1 X PBS, 15% Glycerol, 0.05% Tween-80) and stored frozen at -20°C in aliquots for use in spiking experimental samples.

**Preparation of spiked tongue swabs.** Spiked tongue swabs were prepared by brushing the tongues of healthy volunteers at the University of Washington with Copan FLOQSwabs (Brescia, Italy; 520CS01) for 30 seconds while rotating the swab throughout. For positive controls, the swab head was spiked with MTB cells or MTB gDNA (ATCC) and then broken at its breakpoint for transfer into a 2-mL screwcap-tube. Swabs were stored frozen at -80°C and thawed at room temperature prior to use in experiments. To each thawed sample, 500 μL lysis buffer (10 mM Tris HCl [pH 8], 0.1 mM EDTA [pH 8], 1% Triton-X-100) was added and tubes were heated to 100°C on a heat block for 10 min to kill pathogens and inactivate nucleases. Tubes were allowed to cool to room temperature and then vortexed for 30 seconds to fully release MTB cells from the swab head prior to sonication lysis. Swab collection for experiments with spiked tongue swabs was performed from 8 June 2021 to 30 April 2022, and participants provided written informed consent.

**Sonication lysis.** Spiked tongue swab samples were lysed using a sonication bath (Branson Ultrasonics, Brookfield, CT, USA; CPX1800) filled to the marked operating line with degassed water. Water was degassed by placing a 2-L flask of water on a hotplate and allowing the water to reach a rolling boil for 10 minutes. Immediately after boiling, the degassed water was sealed in 1-L glass screwcap-bottles and allowed to cool to room temperature overnight. Tongue swab samples in 2-mL screwcap-tubes were fixed on a tube holder and immersed in the sonication bath for 10 min, ensuring that the entire swab head was submerged and that no part of the tube holder was in contact with the water (S1 Fig). The tube holder was rotated 90° every 2.5 min to ensure even lysis efficiency. After lysis, the tubes were removed from the water bath and briefly spun down on a table-top centrifuge.

**Hybridization capture.** Purification and concentration of MTB DNA from oral samples was carried out using a sequence-specific hybridization capture method adapted from Oreskovic and Lutz [16] and depicted in Fig 1B. Prior to hybridization capture, Dynabeads MyOne Streptavidin C1 (Thermo Fisher, Waltham, MA, USA) were washed three times in high salt wash buffer (1 M NaCl, 10 mM Tris-HCl [pH 8], 0.05% Tween-20) on a magnetic stand and resuspended in the same volume of high salt wash buffer. A pair of biotinylated capture probes targeting opposite strands of the IS*6110* target region was premixed in Tris-Low-EDTA (TE) buffer (10 mM Tris HCl [pH 8], 0.1mM EDTA [pH 8]) to a final concentration of 5 μM for each probe (10 μM total probe). Probes were ordered with HPLC purification from Integrated DNA Technologies (IDT, Coralville, IA, USA) and designed with a dual biotin modification at the 5' end and a carbon spacer at the 3' end (Probe 1: 5'-/52-Bio/AAAAAAAAAAAAAAAA AAAAGTAGCAGACCTCACCTATG/3SpC3/-3'; Probe 2: 5'-/52-Bio/AAAAAAAAAAAA AAAAAAAACGTAGGCGTCGGTGA/3SpC3/-3').

To begin hybridization capture, 400 μL of sonication-lysed oral samples were transferred to 1.5-mL DNA LoBind tubes (Eppendorf, Hamburg, Germany). Aliquots of 21 μL of 20 mg/mL proteinase K (final concentration 1mg/mL), 2 μL of 10 μM mixed capture probes (20 pmol total), and 0.5 μL of 1 M $CaCl_2$ (final concentration 1.2 μM) were added to each sample and

incubated for 30 min at 55˚C with 1500 RPM shaking for proteinase K digestion. The heat-shaker was then set to 100˚C and 0 RPM and samples were incubated for 10 min to denature the MTB DNA. To hybridize capture probes to MTB DNA, a mixture of 107 μL of 5M NaCl (final concentration 1M) and 5 μL of 10% Tween-20 (final concentration 0.1%) were added to each tube and incubated at 55˚C with 1500 RPM shaking for 20 min.

Following hybridization, 20 μL of washed magnetic beads were added to each sample and tubes were subjected to end-over-end rotation for 1 hour at room temperature. Samples were then placed on a 1.5-mL tube magnetic stand (Sergi Lab Supplies, Seattle, WA, USA) where beads were collected, and the remaining supernatant was discarded. Beads were washed once with 500 μL high salt wash buffer and once with 500 μL low salt wash buffer (15 mM NaCl, 10 mM Tris-HCl [pH 8]). For each wash, tubes were inverted approximately 20 times, or until beads were fully in suspension, briefly centrifuged, and then placed on the magnetic rack for 1 min before the wash buffer was removed. After the final wash step, any residual wash buffer at the bottom of the tubes was removed using a P20 pipet. Purified MTB DNA was eluted by resuspending beads in 20 μL TE buffer and heating to 95˚C for 5 min. Beads were then collected on the magnetic stand and the eluted DNA was quickly (<10 sec) transferred to a PCR tube.

**Qiagen extraction and ethanol precipitation.** To perform the Qiagen extraction and ethanol precipitation reference method, spiked tongue swabs in 2-mL screwcap-tubes were prepared as described above with the following modifications. After thawing to room temperature, swabs were heated to 100˚C in a water bath for 10 min, followed by the addition of 500 μL TE buffer. Samples were then subjected to DNA-extraction using a QIAamp DNA Mini Kit (Qiagen, Hilden, Germany) and concentrated with ethanol precipitation as previously described [4].

**Quantification by qPCR.** Targeted qPCR for IS*6110* was run on DNA-extracted samples as previously described with the following modifications [9]. The qPCR protocol was performed on a CFX96 Touch thermocycler (BioRad, Hercules, CA, USA) using the SensiFAST Probe No-ROX Kit qPCR master mix (Meridian Bioscience, Cincinnati, OH, USA) according to the manufacturer's protocol. In addition to a no-template negative control, a standard curve of purified MTB DNA was included for conversion of Cq values to DNA quantity (S2 Fig).

## Clinical sample evaluation methods

**Collection of clinical samples and test assignment.** Participants were enrolled in TB clinics in Worcester, South Africa, and sampled within 72 hours of TB treatment initiation [17]. Copan FLOQSwabs were collected as described previously [7], except that swabs were stored dry and frozen at -80˚C prior to analysis. Swab collection was performed early in the morning, at least 30 min after consuming food or drink, or oral hygiene. Three consecutive swab samples were collected from each participant. Participants were in two cohorts with overlapping enrollment periods. Participants in Cohort 1 were confirmed TB-positive by sputum Xpert Ultra prior to enrollment. Participants in Cohort 2 had symptoms consistent with possible TB but were tested by sputum Xpert Ultra and culture after enrollment. All samples negative by sputum Xpert Ultra included in this study were therefore from Cohort 2. Detailed definitions of Cohorts 1 and 2, including participant baseline characteristics and inclusion and exclusion criteria, are provided in Wood et al. [17].

Following collection, tongue swabs were frozen dry in 2-mL screwcap-tubes at -80˚C and shipped on dry ice to the University of Washington in Seattle for processing. In the first round of analysis, 30 participants from Cohort 1 with positive sputum Xpert Ultra results were randomly selected for side-by-side processing of hybridization capture or Qiagen extraction and

ethanol precipitation. Two of the three available samples per participant were randomly assigned for paired testing. In the second round of analysis, the remaining 64 participants determined to be positive by sputum Xpert Ultra in Cohort 1 and 42 participants negative by sputum Xpert Ultra and culture in Cohort 2 were assigned only one random swab for processing by hybridization capture for evaluation of sensitivity and specificity, respectively (Fig 2). The N = 94 samples from Cohort 1 include both the 30 participants analyzed in the first round and the 64 participants analyzed in the second round. The study was conducted as a demonstration study, wherein the TB statuses of participants who provided the swab samples were known prior to performing the assays.

**DNA extraction from clinical samples by hybridization capture.** Clinical samples were processed by hybridization capture using the methods described above for spiked samples with a few minor modifications. Frozen dry swabs in 2-mL screwcap-tubes were processed immediately upon thawing. Samples were first heated to 100˚C in a water bath for 10 min for microbiological safety, followed by the addition of 500 µL lysis buffer. Samples were then lysed by sonication and processed by hybridization capture using the methods described above. Each sample processing batch also contained one positive and one negative control, consisting of an MTB-spiked and unspiked tongue swab collected from healthy volunteers at the University of Washington.

**Fig 2. Study flow chart of participants from Cohort 1 and 2 evaluated by hybridization capture.** Sputum Xpert+ = sputum Xpert Ultra-positive; Sputum Xpert− = sputum Xpert Ultra-negative; Cx− = culture-negative.

**DNA extraction from clinical samples by Qiagen extraction and ethanol precipitation.** Clinical swabs were processed by Qiagen extraction and ethanol precipitation using the methods described above for spiked samples. Each sample processing batch also contained one positive and one negative extraction control, consisting of an MTB-spiked and unspiked tongue swab collected from healthy volunteers at the University of Washington.

**Clinical sample analysis by qPCR.** Targeted qPCR for IS*6110* was run on DNA-extracted samples as previously described with the following modifications [9]. For samples that were processed by hybridization capture, 8 μL of eluate were added to 17 μL PCR master mix. For samples processed by Qiagen extraction and ethanol precipitation, dry pellets were resuspended in 8 μL TE and added to 17 μL PCR master mix. Each PCR plate contained a standard curve of purified MTB DNA run in parallel with at least one non-template PCR control. All qPCR exports were normalized to a threshold of 0.2 Δ Rn. Samples were identified as positive if any amplification above the threshold was detected before 45 cycles.

**Ethical considerations.** Data from participant samples were accessed for research purposes from 1 May 2022 to 31 August 2022. Researchers from the University of Washington did not have access to identifiable information about participants. Researchers from the University of Cape Town had access to participant identifiable information both during and after data collection. This study was reviewed and approved by the University of Cape Town Human Research Ethics Committee (reference number 160/2020) and the University of Washington Human Subjects Division (STUDY00001840).

# Results

## MTB DNA extraction efficiency from TE buffer and oral matrix

The impact of sample matrix on DNA yield was evaluated by performing hybridization capture on samples of TE buffer or oral (tongue dorsum) matrix collected by tongue swabs and spiked with $10^3$ copies of MTB DNA. In TE buffer, hybridization capture demonstrated a DNA recovery of 78.7 ± 7.9%. However, when applied to oral matrix samples, the recovery significantly decreased to 36.5 ± 0.8% (Fig 3). Notably, magnetic beads aggregated during processing of oral matrix samples, potentially due to non-specific protein adsorption onto the beads. To address this issue, a proteinase K digestion step was introduced, resulting in a notable improvement of DNA recovery to 89.1 ± 2.7% in oral matrix samples. Consequently, the hybridization capture protocol was modified to include a proteinase K digestion step, as described above, in all future experiments.

## Sonication improves MTB DNA extraction from MTB-spiked tongue swabs

Since mycobacteria, including MTB, have a uniquely robust cell wall, we investigated the potential of sonication to enhance DNA yield by improving lysis efficiency and liberating more DNA for hybridization capture. Tongue swabs spiked with $10^3$ MTB cells were subjected to hybridization capture both with and without upstream sonication. In the absence of sonication, DNA yield from hybridization capture was low at 17 ± 4 pg/rxn, indicating that heat and proteinase K activity alone were insufficient for complete release of MTB DNA (Fig 4). This yield represented only 18% of the MTB DNA recovered from samples that underwent sonication but did not undergo hybridization capture (94 ± 14 pg/rxn). However, the incorporation of sonication prior to hybridization capture significantly improved DNA yield to 827 ± 69 pg/rxn. As a result, a sonication lysis step was implemented into the hybridization capture protocol for all subsequent experiments. Henceforth, we refer to "hybridization capture" as the DNA capture step with the upstream sonication step.

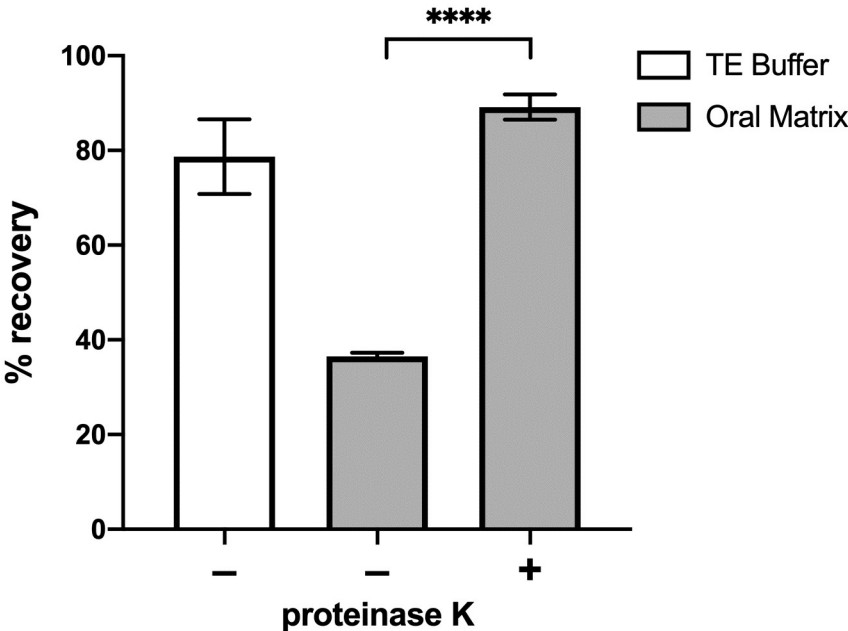

**Fig 3. Addition of proteinase K digestion improves DNA recovery in oral matrix.** Without proteinase K digestion, the DNA recovery of hybridization capture in TE buffer samples was over two-fold higher compared to the recovery in oral matrix samples. However, incorporating a proteinase K digestion step significantly increased the DNA yield from oral matrix samples, reaching levels comparable to that of TE buffer samples (mean ± SD, n = 3). **** indicates P value of < 0.0001 (two-sample t-test).

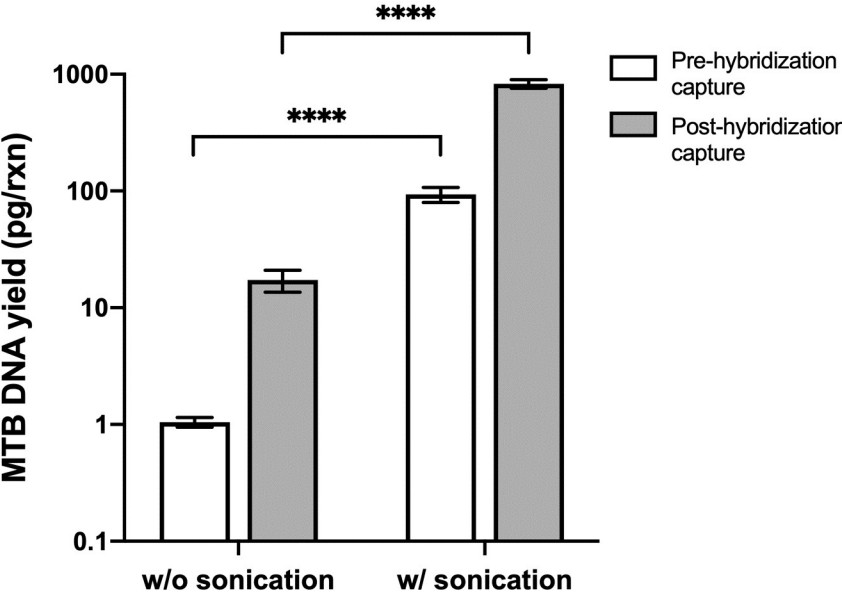

**Fig 4. Sonication improves DNA yield from MTB-spiked tongue swabs.** Hybridization capture alone demonstrated limited efficiency in extracting MTB DNA from MTB cells without mechanical lysis. However, the inclusion of a sonication lysis step resulted in a significant enhancement of DNA yield both prior to and after the hybridization capture step. Data is presented on a logarithmic scale (mean ± SD, n = 3). **** indicates P value of < 0.0001 (two-sample t-test).

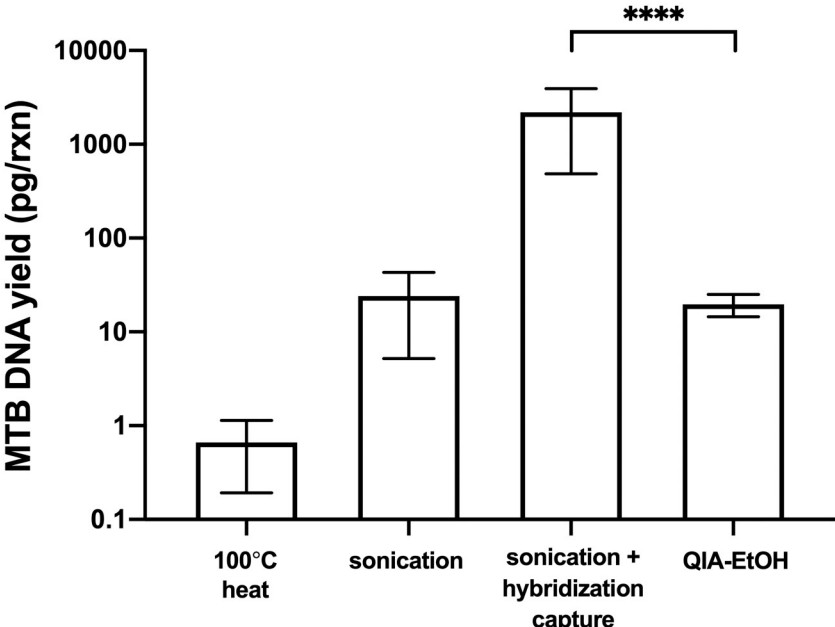

**Fig 5. Hybridization capture improves DNA yield over QIA-EtOH in MTB-spiked samples.** Hybridization capture resulted in a substantial (~100-fold) increase in MTB DNA compared to the historically used QIA-EtOH method when applied to tongue swabs spiked with $10^4$ MTB cells. MTB DNA yield was also measured following the 100˚C boil step and the sonication lysis step as part of the hybridization capture process. Data is presented on logarithmic scale (mean ± SD, n = 8). **** indicates P value of < 0.0001 (two-sample t-test).

## Comparison of hybridization capture to Qiagen DNA extraction and ethanol precipitation in MTB-spiked tongue swabs

We assessed whether DNA extraction was improved by hybridization capture, relative to previous methods, which involved using a Qiagen DNA extraction kit followed by ethanol precipitation (QIA-EtOH). When applied to tongue swabs spiked with $10^4$ MTB cells, hybridization capture was significantly more effective, with an average DNA yield of 2200 ± 1600 pg/rxn, compared to QIA-EtOH which yielded only 20 ± 5 pg/rxn (Fig 5). Notably, the DNA yield from QIA-EtOH was not significantly different from that obtained with sonication alone (24 ± 17 pg/rxn). DNA yield from 100˚C heat alone was 0.7 ± 0.4 pg/rxn. Swabs were spiked with $10^4$ MTB cells (as opposed to $10^3$ previously) because we anticipated poor performance from QIA-EtOH, and we aimed to quantify samples by qPCR in this group.

## Comparison of sensitivity between hybridization capture and QIA-EtOH in clinical tongue swabs

The sensitivity of hybridization capture was compared to that of QIA-EtOH using paired clinical samples collected from 30 sputum Xpert Ultra-positive study participants from Cohort 1. For each participant, two swabs were randomly selected from three consecutively collected swabs, and each selected swab was processed using one of the two methods. The sensitivity of hybridization capture was 77% (23/30), while the sensitivity of QIA-EtOH was 73% (22/30). Among the 30 pairs of swabs analyzed, 21 pairs were identified as true positives by both hybridization capture and QIA-EtOH. Hybridization capture detected two true positives that were missed by QIA-EtOH, while QIA-EtOH detected one true positive that was missed by hybridization capture.

### Sensitivity and specificity of hybridization capture in clinical tongue swabs

To further evaluate the sensitivity and determine the specificity of hybridization capture, we applied this method to an additional 64 swabs collected from sputum Xpert Ultra TB-positive South Africans to completely test Cohort 1 (94 TB-positive swabs total) and 42 swabs from sputum Xpert Ultra and culture TB-negative South Africans from Cohort 2. The overall sensitivity of hybridization capture was 74% (70/94) and specificity was 98% (41/42).

## Discussion

The use of tongue swabs offers several advantages for TB screening, particularly for individuals with limited sputum production or in settings where sputum collection is impractical. To maximize the accuracy of tongue swab testing relative to sputum testing, efficient DNA extraction and concentration methods are crucial. Previous studies have employed commercially available Qiagen spin columns for DNA extraction and ethanol precipitation for sample concentration [4–9]. Although these studies demonstrated the potential of tongue swabs in detecting MTB DNA, the reported sensitivities and specificities remained lower than those achieved by sputum-based assays, and these methods were labor-intensive. We hypothesized that the simplicity and sequence-specific nature of our hybridization capture method could improve diagnostic accuracy and ease of use compared to QIA-EtOH.

In this study, we highlight two elements of our hybridization capture method to optimize performance in tongue swabs: proteinase K digestion and mechanical lysis. The inclusion of a proteinase K digestion step is essential for mitigating interference caused by protein derived from the oral cavity (Fig 3). In the absence of proteinase K digestion, we observed bead aggregation due to the adsorption of heat-denatured protein onto the bead surface [18]. This aggregation may inhibit DNA capture by reducing the available functional surface area and causing inefficient washing and elution.

We also found that the inclusion of a sonication lysis step enhances DNA recovery from MTB-spiked tongue swabs. MTB is resistant to conventional methods of bacterial lysis due to unique structures in its cell wall, including long-chain mycolic acids and various polysaccharides [12]. Sonication was selected as the lysis method in this study due to its proven efficacy in MTB lysis, as demonstrated in other studies [19, 20]. Our experiments with MTB-spiked swabs revealed a significant increase in DNA yield from swabs that underwent both sonication lysis and hybridization capture compared to those subjected to hybridization capture alone (Fig 4). These results highlight the importance of a robust mechanical lysis step to facilitate the release of free DNA for subsequent hybridization capture.

As hypothesized, the optimized hybridization capture protocol, which includes proteinase K digestion and sonication, demonstrated a significant enhancement in MTB DNA recovery from MTB-spiked tongue swabs compared to QIA-EtOH (Fig 5). This improvement may be attributed to several factors. First, unlike the silica columns utilized in Qiagen DNA extraction kits, hybridization capture is sequence-specific, reducing the likelihood of non-MTB DNA from the oral flora entering the PCR reaction and inhibiting amplification [14]. Additionally, in the presence of high concentrations of non-MTB DNA, the efficiency of MTB DNA binding to silica columns may decrease due to competition for the column's limited binding capacity. Finally, mechanical lysis may free more DNA to be available for capture compared to the gentler enzymatic and chemical lysis methods used in QIA-EtOH. We found that sonication prior to Qiagen DNA extraction yielded a 100-fold increase in DNA yield over samples that underwent Qiagen DNA extraction alone (S3 Fig).

Despite improvements in spiked tongue swabs, there was no significant difference in sensitivity between hybridization capture (77%) and QIA-EtOH (73%) in 30 pairs of clinical

samples from Xpert Ultra-positive participants. The overall sensitivity of hybridization capture did not improve when applied to an additional 64 clinical samples (74%, 70/94). MTB cells from clinical tongue swabs may be more fragile due to prolonged exposure to salivary enzymes and oral flora compared to the cultured H37Ra MTB cells used to optimize hybridization capture. Consequently, the enhanced sensitivity provided by sonication may have had a relatively smaller impact on clinical MTB cells, as they might already be susceptible to the gentle lysis methods used in QIA-EtOH. Complete lysis of MTB cells in clinical swabs is unlikely, as studies have successfully amplified MTB DNA from tongue swabs using the Xpert Ultra, which filters and amplifies intact MTB cells [10]. Wood et al. [7] also demonstrated the recovery of viable MTB cells from tongue swab samples by bacteriological culture.

Limitations of this study may also contribute to a greater proportion of fragile MTB cells and/or cell-free DNA in clinical samples. These include the collection of samples post-TB treatment (up to 72 hours) and the processing of samples following a freeze-thaw cycle, which is known to disrupt bacterial integrity [21]. Our study is also limited in that culture data was not collected for Xpert Ultra-positive participants. While QIA-EtOH was tested on a small sample size of 30, our study is strengthened by testing two paired swabs per participant, one for each method.

Another potential factor influencing sensitivity in this particular set of clinical swabs is the high concentration of biotin present in oral samples, which may compete with the biotinylated capture probes for binding to the streptavidin-coated magnetic beads. Biotin is primarily absorbed from food, and it is possible that swabs collected following the consumption of a biotin-rich meal may lead to a false negative result. This study is also limited in applicability to other settings and populations. Evaluation of other cohorts is warranted.

Although hybridization capture did not exhibit higher sensitivity in clinical swabs compared to QIA-EtOH, its specificity in this evaluation (98%) surpassed that observed in past studies utilizing the more laborious QIA-EtOH method (78–93%) [5–8]. We hypothesize that the ethanol precipitation step in QIA-EtOH poses a substantial risk for cross-contamination, as the tubes remain open for an extended period during this step. In contrast, hybridization capture mitigates such risks and involves fewer overall pipetting steps. Notably, this improved performance was achieved without a significant difference in total reagent cost ($4.50 for hybridization capture vs. $4.00 for QIA-EtOH at the time of publishing).

Going forward, the sensitivity of the methods described here can be further improved by targeting additional multi-copy sequences in the MTB genome, such as IS*1081*. Unlike IS*6110*, which can range from 0–25 copies per genome, IS*1081* is present in all MTB strains at a more consistent copy number of 5–6 repeats [22, 23]. Additionally, pre-binding biotinylated capture probes to magnetic beads prior to incubation with oral samples may enhance resistance to free biotin, as demonstrated by Oreskovic and Lutz [16]. Since magnetic bead-based nucleic acid capture is compatible with many automated platforms [24, 25] there is also potential to improve ease of use.

In conclusion, our results indicate that hybridization capture is an effective sample preparation strategy for tongue swabs due to its ease of use and high specificity relative to traditional DNA extraction methods. With further optimization to improve sensitivity, this method can help tongue swab testing become a feasible alternative to sputum-based testing and expand access to rapid TB diagnosis to key underserved populations.

## Supporting information

**S1 Fig. Image of sample and tube rack positioning in sonication bath.**
(TIF)

**S2 Fig. Representative standard curve for conversion of Cq values to DNA quantity.** Representative calibration curve illustrating qPCR standards across a range of concentrations from 0.01–1000 pg/µL (0.05–5000 pg/rxn) of purified MTB DNA (mean ± SD, n = 7).
(TIF)

**S3 Fig. Sonication lysis upstream of Qiagen extraction increases DNA recovery.** The QIAamp DNA extraction kit was performed on tongue swabs spiked with $10^4$ H37Ra MTB cells both with and without an upstream sonication lysis step. MTB DNA yield was significantly higher when sonication was included. **** indicates P value of < 0.0001 (two-sample t-test). Data is presented on a logarithmic scale (mean ± SD, n = 3).
(TIF)

**S4 Fig. Triton-X-100 does not affect DNA recovery for hybridization capture.** Hybridization capture was performed on tongue swabs spiked with $10^3$ H37Ra MTB cells and resuspended in either TE buffer or lysis buffer (TE, 1% Triton-X-100). There was no significant difference in MTB DNA yield between the two groups (mean ± SD, n = 3). NS indicates P-value > 0.05 (two-sample t-test).
(TIF)

**S1 File. Source table containing Cq values and DNA quantification data from clinical sample qPCR tests.**
(CSV)

# Acknowledgments

We thank our colleagues Dr. Amy Oreskovic and Prof. Barry Lutz of the UW Department of Bioengineering for their input on hybridization capture, as well as Prof. Adam Maxwell of the UW Department of Urology for his input on the sonication aspect of this work. We also express our gratitude to the members of the Yager group and the Cangelosi group, especially Rachel Wood, for their insightful comments throughout the project.

# Author Contributions

**Conceptualization:** Alexander J. Yan, Kris M. Weigel, Gerard A. Cangelosi, Paul Yager.

**Data curation:** Alaina M. Olson.

**Formal analysis:** Alexander J. Yan, Alaina M. Olson, Kris M. Weigel.

**Funding acquisition:** Alexander J. Yan, Gerard A. Cangelosi, Paul Yager.

**Investigation:** Alexander J. Yan, Alaina M. Olson, Angelique K. Luabeya.

**Methodology:** Alexander J. Yan, Alaina M. Olson, Kris M. Weigel, Erin Heiniger, Gerard A. Cangelosi, Paul Yager.

**Project administration:** Kris M. Weigel, Angelique K. Luabeya, Mark Hatherill, Gerard A. Cangelosi, Paul Yager.

**Resources:** Angelique K. Luabeya, Gerard A. Cangelosi, Paul Yager.

**Supervision:** Kris M. Weigel, Angelique K. Luabeya, Mark Hatherill, Gerard A. Cangelosi, Paul Yager.

**Validation:** Alexander J. Yan, Alaina M. Olson, Kris M. Weigel.

**Visualization:** Alexander J. Yan.

**Writing – original draft:** Alexander J. Yan, Alaina M. Olson.

**Writing – review & editing:** Alexander J. Yan, Alaina M. Olson, Kris M. Weigel, Angelique K. Luabeya, Erin Heiniger, Mark Hatherill, Gerard A. Cangelosi, Paul Yager.

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
