## [Decision Letter · Decision Letter 0]

14 Dec 2023

PONE-D-23-25589Detection of *Mycobacterium tuberculosis* from tongue swabs using sonication and sequence-specific hybridization capture**PLOS ONE

Dear Dr. Yan,

Thank you for submitting your manuscript to PLOS ONE. After careful consideration, we feel that it has merit but does not fully meet PLOS ONE’s publication criteria as it currently stands. Therefore, we invite you to submit a revised version of the manuscript that addresses the points raised during the review process.

We look forward to receiving your revised manuscript.

Kind regards,

Atul Vashist, PhD

Academic Editor

PLOS ONE

Journal Requirements:

"I have read the journal's policy and the authors of this manuscript have the following competing interests: Paul Yager has a nonpaying appointment as CSO of the UbiDX corporation. There are no patents, products in development, or marketed products associated with this research to declare. Swabs were kindly donated by Copan"

*Reviewers' comments:*

*Reviewer's Responses to Questions*

*

**Comments to the Author**
*

1. Is the manuscript technically sound, and do the data support the conclusions?

*The manuscript must describe a technically sound piece of scientific research with data that supports the conclusions. Experiments must have been conducted rigorously, with appropriate controls, replication, and sample sizes. The conclusions must be drawn appropriately based on the data presented. *

*Reviewer #1: Partly*

*Reviewer #2: Yes*

*Reviewer #3: Yes*

*2. Has the statistical analysis been performed appropriately and rigorously? *

*Reviewer #1: Yes*

*Reviewer #2: No*

*Reviewer #3: Yes*

*3. Have the authors made all data underlying the findings in their manuscript fully available?*

*The PLOS Data policy requires authors to make all data underlying the findings described in their manuscript fully available without restriction, with rare exception (please refer to the Data Availability Statement in the manuscript PDF file). The data should be provided as part of the manuscript or its supporting information, or deposited to a public repository. For example, in addition to summary statistics, the data points behind means, medians and variance measures should be available. If there are restrictions on publicly sharing data—e.g. participant privacy or use of data from a third party—those must be specified.*

*Reviewer #1: No*

*Reviewer #2: Yes*

*Reviewer #3: Yes*

*4. Is the manuscript presented in an intelligible fashion and written in standard English?*

*PLOS ONE does not copyedit accepted manuscripts, so the language in submitted articles must be clear, correct, and unambiguous. Any typographical or grammatical errors should be corrected at revision, so please note any specific errors here.*

*Reviewer #1: Yes*

*Reviewer #2: Yes*

*Reviewer #3: Yes*

*5. Review Comments to the Author*

*Please use the space provided to explain your answers to the questions above. You may also include additional comments for the author, including concerns about dual publication, research ethics, or publication ethics. (Please upload your review as an attachment if it exceeds 20,000 characters)*

*Reviewer #1: In ‘Materials and Methods’ section, under the heading ‘Methods for DNA extraction in spiked oral swabs,’ subheading ‘Quantification by qPCR,’ you mentioned a standard curve of purified MTB DNA was used for conversion of Cq values to DNA quantity, but the same was not included in the document. It could provide a reference for the range of DNA quantity versus Cq values you have used in the study.*

In ‘Fig 1. Workflow of MTB DNA extraction from oral swabs,’ it would be beneficial if you could include in the figure the temperature and duration you have used for individual steps

In ‘Materials and Methods’ section, under the heading ‘Clinical sample evaluation methods’, subheading ‘collection of clinical samples and test assignment,’ you mentioned that participants were in two cohorts, however, you did not mention the number of participants in each cohort included and the impact of the two cohorts will have on the conducted study.

In ‘Materials and Methods’ section, for preparation of spiked oral swabs for hybridization capture, swab head was heated to 100 °C for 10 mins in lysis buffer, but for preparation of spiked oral swabs for QIA-EtOH and clinical oral swabs for hybridization capture and QIA-EtOH you heated the swab to 100 °C for 10 min and then TE buffer or lysis buffer was added. Won’t heating in the buffer and heating without the buffer impact the quantity of DNA extracted and the accuracy of the extraction process?

In ‘Results’ section, under the heading ‘Comparison of hybridization capture to Qiagen DNA extraction and ethanol precipitation in MTB-spiked oral swabs,’ the average DNA yield for spiked oral swabs by hybridization capture was mentioned as 2200 ± 1600 pg/rxn. A variation of ± 1600 pg is very large, you did not mention the reason for such variation.

Also, for sample matrix effect and sonication effect measurement you used Oral swabs spiked with 103 MTB cells but for Comparison of hybridization capture to QIA-EtOH DNA extraction you used oral swabs spiked with 104 MTB cells. The document does not include the reason for this increase in the number of MTB cells for spiking.

In the ‘Fig 4. Hybridization capture improves DNA yield over QIA-EtOH in MTB-spiked samples,’ the bar for sonication alone was drawn bigger than the value it was representing 27 ± 24 pg/rxn and the DNA yield value for 100°C boil is not included in the figure description.

MTB cell concentration estimation for the clinical samples is not mentioned in the document, such detail could be useful to know the extent to which the optimized method used in this study for oral swabs spiked with 103 or 104 MTB cells holds good when clinical samples were considered.

In ‘Results’ section, under the heading ‘Comparison of sensitivity between hybridization capture and QIA-EtOH in clinical oral swabs,’ the specificity values of hybridization capture and QIA-EtOH is not included, which could be helpful in comparing the false positive results.

Also, why you did not consider all the 94 sputum Xpert Ultra-positive study participants altogether instead of considering only 30 of them, for the comparison of sensitivity between hybridization capture and QIA-EtOH in clinical oral swabs, since, on increasing the number of participants, the sensitivity for the hybridization capture was decreasing, then how to confirm that similar trend is also observed in the QIA-EtOH method in clinical oral swabs.

For the preparation of spiked oral swabs, you used lysis buffer containing 1% Triton-X-100 detergent. As Triton-X-100 itself is a lysing agent, it would be useful if you also include some experimental data representing the comparison of hybridization capture to QIA-EtOH in MTB-spiked oral swabs with and without 1% Triton-X-100 for the efficiency of the process on the DNA yield.

Under ‘Fig 4.’ description it was given that sonication with hybridization capture resulted in a ~100-fold increase in DNA yield compared to QIA-EtOH extraction alone. Similarly, under ‘discussion’ section it was given that sonication prior to Qiagen DNA extraction yielded a 100-fold increase in DNA yield compared to QIA-EtOH extraction alone. If sonication showing 100-fold increase in DNA yield with both Hybridization capture and with QIA-EtOH, then how your developed method is better than Qiagen apart from cross contamination protection, since, it would be much easier to use sonication with QIA-EtOH then sonication with hybridization capture which requires designing of probes for specific target sequences.

*It would be useful if you include some cost comparison between the above two methods.*

*Reviewer #2: The authors in this study entitled “Detection of Mycobacterium tuberculosis from tongue swabs using sonication and sequence-specific hybridization capture” try to elucidate the utility of oral swabs as a non-invasive sample for diagnosing tuberculosis. The study is interesting and I have a few comments as below*

• The authors have mentioned several methods and modifications (proteinase K, sonication, hybridization capture etc) which have improved the DNA isolation from oral swabs. However (in all experiments Fig 2, 3 and 4), no statistics has been applied to ascertain whether this improvement was significant or not? How many samples were included in these experiments? Three? This is especially important as the same difference was not seen when applied to real samples and not spiked samples.

• Line 246-247. “Samples were identified as positive if any amplification (above the threshold) was detected after 45 cycles”. Please clarify this. For qPCR any amplification (above the threshold) was detected after 45 cycles?

• Pg 9. Please include a workflow for the whole study in accordance with the STARD guidelines defining the Cohort 1 and cohort 2. The protocols can also be included in work flow format.

• How was sensitivity and specificity calculated? Xpert Ultra as gold standard? I am unable to align the results obtained with the supplementary data provided. Were the results of both the Cohorts combined for analysis? Was culture data also compared for analysis?

• Please mention that study was done as a demonstration study, wherein results of the samples were known prior to performing the assay.

*• The authors should shorten the discussion and mention strengths and limitations of the study.*

*Reviewer #3: The study detection of Mtb from tongue swabs using sonication and sequence based hybridization capture is a need of the time to achieve the goal of end TB. the study design and subject is very good but have some limitations like:*

1. the authors didnt provide the clinical details of the samples included in the study for evaluation of the samples.

2. to evaluate the specificity the authors needs to also include the sample from similar diseased control of non-TB patients to check the cross reactivity.

3. the formatting of manuscript is poorly done and in between the paragraph/test figure and table legends inserted.

4. need revision of minor spelling corrections in the manuscript.

*5. Authors should include the table for clinical details of samples tested and mentioning the sensitivity and specificity of tested samples for better representation of data.*

*6. PLOS authors have the option to publish the peer review history of their article (what does this mean?). If published, this will include your full peer review and any attached files.*

**

*Reviewer #1: No*

*Reviewer #2: No*

*Reviewer #3: No*

**

*While revising your submission, please upload your figure files to the Preflight Analysis and Conversion Engine (PACE) digital diagnostic tool, https://pacev2.apexcovantage.com/. PACE helps ensure that figures meet PLOS requirements. To use PACE, you must first register as a user. Registration is free. Then, login and navigate to the UPLOAD tab, where you will find detailed instructions on how to use the tool. If you encounter any issues or have any questions when using PACE, please email PLOS at figures@plos.org. Please note that Supporting Information files do not need this step.*

---

## [Author Response · Author response to Decision Letter 0]

30 Jan 2024

Response to Reviewers

Dear PLOS ONE reviewers,

Thank you for the opportunity to resubmit our manuscript titled “Detection of Mycobacterium tuberculosis from tongue swabs using sonication and sequence-specific hybridization capture” for publication in PLOS ONE. We appreciate the time and effort dedicated to providing feedback on our manuscript, and we have made the following changes to our submission to address the feedback provided. These changes are found in the revised manuscript file with tracked changes. Please see below, in blue, for a point-by-point response to the reviewers’ comments and concerns. All line numbers refer to the clean revised manuscript file without tracked changes. 

Reviewers' Comments to the Authors:

Reviewer # 1:

Reviewer comment 1: In ‘Materials and Methods’ section, under the heading ‘Methods for DNA extraction in spiked oral swabs,’ subheading ‘Quantification by qPCR,’ you mentioned a standard curve of purified MTB DNA was used for conversion of Cq values to DNA quantity, but the same was not included in the document. It could provide a reference for the range of DNA quantity versus Cq values you have used in the study.

Author response: Throughout our study, multiple qPCR reactions were performed, each accompanied by its own standard curve. Due to space limitations and to avoid redundancy, we have chosen to include one representative standard curve (S2 Fig) as a reference for the reader. We believe this new supplemental figure adequately captures the range of the DNA quantity used to create the standard curves in this study. We appreciate this helpful suggestion.

Reviewer comment 2: In ‘Fig 1. Workflow of MTB DNA extraction from oral swabs,’ it would be beneficial if you could include in the figure the temperature and duration you have used for individual steps

Author response: We appreciate your suggestion regarding the addition of temperature and duration information for individual steps in the figure. However, we intentionally omitted this information from the figure to maintain clarity and prevent visual clutter. Our objective with this figure was to provide a concise overview of the main steps in the hybridization capture protocol. For readers who want to see the detailed temperature and duration for each step, we have included this information in the figure description of Fig 1 (lines 118–124). 

Reviewer comment 3: In ‘Materials and Methods’ section, under the heading ‘Clinical sample evaluation methods’, subheading ‘collection of clinical samples and test assignment,’ you mentioned that participants were in two cohorts, however, you did not mention the number of participants in each cohort included and the impact of the two cohorts will have on the conducted study.

Author response: In response to the suggestion from one of the reviewers, we have put together a STARD diagram (Fig 2) that should clarify to the readers the number of participants in each cohort and how samples from each cohort were used. Changes to the text have been made to clarify which cohorts were involved in the sensitivity and specificity calculations, including the addition of lines 220–221, which now states: “All negative by Xpert Ultra samples included in this study are therefore from Cohort 2.”

Regarding the impact on the study, the study was not constructed to compare Cohort 2’s participants, who were initially presumptive TB cases but later confirmed negative for TB, to Cohort 1’s participants, who were all confirmed-TB cases. Both cohorts share the same enrollment sites, time periods partially overlapped, and participants were all voluntarily healthcare seeking. Due to Cohort 1’s structure of collection and enrollment (designed for a primary study different to this one) we would not have had any clinically negative confirmed samples to test. The inclusion of negative samples from Cohort 2 enhances the robustness of our study. No samples confirmed positive for TB from Cohort 2 were included in the analyses. 

Reviewer comment 4: In ‘Materials and Methods’ section, for preparation of spiked oral swabs for hybridization capture, swab head was heated to 100 °C for 10 mins in lysis buffer, but for preparation of spiked oral swabs for QIA-EtOH and clinical oral swabs for hybridization capture and QIA-EtOH you heated the swab to 100 °C for 10 min and then TE buffer or lysis buffer was added. Won’t heating in the buffer and heating without the buffer impact the quantity of DNA extracted and the accuracy of the extraction process?

Author response: The hybridization capture protocol was optimized for heating the swab in lysis buffer because we initially hypothesized that this would improve DNA yield. However, subsequent experimentation revealed that there was no significant difference in DNA yield between heating in buffer and heating without buffer when applied to spiked samples (data not shown here). We speculate that any potential drawbacks associated with heating the swabs without buffer were mitigated by the robust lysis performance of sonication. 

In the case of experiments conducted on spiked samples using QIA-EtOH, we followed the established protocol used by our consortium (e.g. Luabeya et al, 2019 [1]). However, the clinical swabs we received were stored dry and required an inactivation step (heating to 100 °C) before opening for biosafety reasons. Therefore, we performed the heating step without buffer in these samples. Although the modification to heat the swabs without buffer in the clinical samples may have affected DNA extraction efficiency, biosafety concerns overrode this issue. 

Reviewer comment 5: In ‘Results’ section, under the heading ‘Comparison of hybridization capture to Qiagen DNA extraction and ethanol precipitation in MTB-spiked oral swabs,’ the average DNA yield for spiked oral swabs by hybridization capture was mentioned as 2200 ± 1600 pg/rxn. A variation of ± 1600 pg is very large, you did not mention the reason for such variation.

Author response: While we respectfully acknowledge the concern about this high standard deviation, we believe it is important to clarify that this variation is within the expected range based on our experimental conditions. 

Various sources of error may contribute to this observed variation, as discussed in the manuscript. Notably, donor swab variability plays a significant role. As mentioned in lines 413–417, high concentration of biotin in certain swabs may compete with biotinylated capture probes for binding to magnetic beads, leading to decrease in DNA yield. Furthermore, other factors including presence of food particles, nucleases, or other inhibitors in swab samples could lead to variable DNA recovery. 

Reviewer comment 6: Also, for sample matrix effect and sonication effect measurement you used Oral swabs spiked with 103 MTB cells but for Comparison of hybridization capture to QIA-EtOH DNA extraction you used oral swabs spiked with 104 MTB cells. The document does not include the reason for this increase in the number of MTB cells for spiking.

Author response: In our experiments investigating the effects of sample matrix and sonication, we intentionally spiked samples with 103 MTB cells. This lower cell count was selected to better replicate the lower quantity of MTB cells expected to be found on many clinical swabs. 

In contrast, when comparing hybridization capture to QIA-EtOH, we increased the spiked cell count to 104 MTB cells. This decision was influenced by our anticipation, derived from prior experiments, that QIA-EtOH would perform sub-optimally in spiked swabs. The higher cell count in this comparison was chosen to minimize the risk of qPCR drop-offs for these specific samples. We believe that these variations in spiking quantity were crucial to addressing the specific objectives of each experimental set. To provide clarity on this for the reader, we have included this reasoning in the manuscript (lines 323–325) which states: “Swabs were spiked with 104 MTB cells (as opposed to 103 previously) because we anticipated poor performance from QIA-EtOH, and we aimed to quantify samples by qPCR in this group.”

Reviewer comment 7: In the ‘Fig 4. Hybridization capture improves DNA yield over QIA-EtOH in MTB-spiked samples,’ the bar for sonication alone was drawn bigger than the value it was representing 27 ± 24 pg/rxn and the DNA yield value for 100°C boil is not included in the figure description.

Author response: Thank you for bringing this to our attention. Upon further examination, we discovered that in Fig 4 (now renamed to Fig 5) the bar for “sonication alone” was indeed drawn inaccurately due to inputting the incorrect value in the graph-making software. Upon further inspection, we also found that the stated value of 27 ± 24 pg/rxn was incorrect, and the correct value should have been 24 ± 17 pg/rxn. These corrections have been made in the manuscript (line 322) and in the figure itself. This revised value now accurately represents the size of the bar for “sonication alone”. We apologize for any confusion caused by this discrepancy. Please note that this figure has now been renumbered to Fig 5.

Furthermore, we have included the DNA yield value for the 100 °C boil condition (line 322). 

Reviewer comment 8: MTB cell concentration estimation for the clinical samples is not mentioned in the document, such detail could be useful to know the extent to which the optimized method used in this study for oral swabs spiked with 103 or 104 MTB cells holds good when clinical samples were considered.

Author response: It was not possible to predict MTB cell concentrations in swabs prior to analysis. However, the values of 103 to 104 are consistent with measurements recently reported by Amy Steadman et al [2], Figure 1. While exact cell counts cannot be quantified, we would like to highlight that DNA quantities for each clinical swab are reported in the supplemental materials (S1 File).

Reviewer comment 9: In ‘Results’ section, under the heading ‘Comparison of sensitivity between hybridization capture and QIA-EtOH in clinical oral swabs,’ the specificity values of hybridization capture and QIA-EtOH is not included, which could be helpful in comparing the false positive results.

Author response: In this particular investigation, QIA-EtOH was not used to process any negative clinical samples. However, our consortium had previously conducted numerous studies (referenced at line 422) specifically aimed at characterizing the specificity of QIA-EtOH. Our focus in this paper was on characterizing the specificity of our hybridization capture method, which we observed to be an improvement over past specificity values obtained with QIA-EtOH. 

Reviewer comment 10: Also, why you did not consider all the 94 sputum Xpert Ultra-positive study participants altogether instead of considering only 30 of them, for the comparison of sensitivity between hybridization capture and QIA-EtOH in clinical oral swabs, since, on increasing the number of participants, the sensitivity for the hybridization capture was decreasing, then how to confirm that similar trend is also observed in the QIA-EtOH method in clinical oral swabs.

Author response: Because clinical samples are a limited resource, it can be problematic to expend them on experiments involving previous methods. Therefore, the sensitivity comparison between hybridization capture and QIA-EtOH was limited to a subset of 30 samples. The primary goal of this subset comparison was to validate the compatibility of hybridization capture with clinical samples. Once this compatibility was confirmed, we extended the sensitivity analysis of hybridization capture to include the remaining 64 sputum Xpert Ultra-positive samples. Our consortium had already undertaken extensive studies (described in lines 71–75) characterizing the sensitivity of QIA-EtOH. Thus, we deliberately chose to allocate our time and resources to comprehensively characterize the sensitivity of hybridization capture. 

Reviewer comment 11: For the preparation of spiked oral swabs, you used lysis buffer containing 1% Triton-X-100 detergent. As Triton-X-100 itself is a lysing agent, it would be useful if you also include some experimental data representing the comparison of hybridization capture to QIA-EtOH in MTB-spiked oral swabs with and without 1% Triton-X-100 for the efficiency of the process on the DNA yield.

Author response: We appreciate this suggestion. We have previously performed experiments comparing the DNA yield of hybridization capture in MTB-spiked oral swabs with and without 1% Triton-X-100. In these experiments we did not observe any difference in DNA yield between these two conditions. We speculate that the robustness of sonication may have obviated the need for the additional lysing power of Triton-X-100 for effective MTB lysis. To convey this information to the reader, we have included the corresponding data as a supplementary figure (S4 Fig). 

Reviewer comment 12: Under ‘Fig 4.’ description it was given that sonication with hybridization capture resulted in a ~100-fold increase in DNA yield compared to QIA-EtOH extraction alone. Similarly, under ‘discussion’ section it was given that sonication prior to Qiagen DNA extraction yielded a 100-fold increase in DNA yield compared to QIA-EtOH extraction alone. If sonication showing 100-fold increase in DNA yield with both Hybridization capture and with QIA-EtOH, then how your developed method is better than Qiagen apart from cross contamination protection, since, it would be much easier to use sonication with QIA-EtOH then sonication with hybridization capture which requires designing of probes for specific target sequences.

Author response: We appreciate this observation. In addition to being less susceptible to cross-contamination, hybridization capture is in fact easier than QIA-EtOH when targeting a specific known pathogen. 

As referenced in lines 434–436, hybridization capture is an established technique that is used in automated platforms for nucleic acid capture/amplification, including Nuclein’s DASH platform. Automation is not possible with EtOH precipitation. Therefore, we consider our hybridization capture protocol for oral swabs as a proof-of-principle and a stepping-stone towards automated approaches using this methodology. 

With regard to the need to design probes, it needs to be done only once for each pathogen. Once it is reported here, it does not need to be done again for M. tuberculosis. 

Reviewer comment 13: It would be useful if you include some cost comparison between the above two methods.

Author response: Thank you for this valuable suggestion. We have included a cost comparison between the two methods in the discussion (lines 426–428), which states: “Notably, this improved performance was achieved without a significance difference in total reagent cost ($4.50 for hybridization capture vs. $4.00 for QIA-EtOH at the time of publishing).”

Reviewer # 2:

Reviewer comment 14: The authors have mentioned several methods and modifications (proteinase K, sonication, hybridization capture etc) which have improved the DNA isolation from oral swabs. However (in all experiments Fig 2, 3 and 4), no statistics has been applied to ascertain whether this improvement was significant or not? How many samples were included in these experiments? Three? This is especially important as the same difference was not seen when applied to real samples and not spiked samples.

Author response: Thank you for bringing up this important point. The number of samples included in these experiments is specified in the figure caption. Three samples were included in the experiments for Fig 2 (renamed to Fig 3) and Fig 3 (renamed to Fig 4), while eight samples were included in experiments for Fig 4 (renamed to Fig 5). These sample numbers can be found in lines 290, 311, and 332 respectively. In response to your suggestion, we have now added brackets with P-values to these figures to visually indicate the statistical significance of each modification. Please note, in this latest submission these figures have been renumbered from “2, 3, and 4” to “3, 4, and 5” respectively. 

Reviewer comment 15: Line 246-247. “Samples were identified as positive if any amplific

---

## [Decision Letter · Decision Letter 1]

24 May 2024

PONE-D-23-25589R1Detection of *Mycobacterium tuberculosis* from tongue swabs using sonication and sequence-specific hybridization capture***PLOS ONE*

*Dear Dr. *Yan,

*Thank you for submitting your manuscript to PLOS ONE. After careful consideration, we feel that it has merit but does not fully meet PLOS ONE’s publication criteria as it currently stands. **Therefore, we invite you to submit a revised version of the manuscript that addresses the points raised by Reviewer 3 during the review process. * *Please submit your revised manuscript by *Jul 08 2024 11:59PM*. If you will need more time than this to complete your revisions, please reply to this message or contact the journal office at plosone@plos.org. *

*Please include the following items when submitting your revised manuscript:*

*A rebuttal letter that responds to each point raised by the academic editor and reviewer(s). You should upload this letter as a separate file labeled 'Response to Reviewers'.*

*A marked-up copy of your manuscript that highlights changes made to the original version. You should upload this as a separate file labeled 'Revised Manuscript with Track Changes'.*

*An unmarked version of your revised paper without tracked changes. You should upload this as a separate file labeled 'Manuscript'.*

**

*We look forward to receiving your revised manuscript.*

*Kind regards,*

*Atul Vashist, PhD*

Academic Editor

*PLOS ONE*

*Journal Requirements:*

Reviewers' comments:

*Reviewer's Responses to Questions*

*

**Comments to the Author**
*

*1. If the authors have adequately addressed your comments raised in a previous round of review and you feel that this manuscript is now acceptable for publication, you may indicate that here to bypass the “Comments to the Author” section, enter your conflict of interest statement in the “Confidential to Editor” section, and submit your "Accept" recommendation.*

*Reviewer #1: All comments have been addressed*

*Reviewer #2: All comments have been addressed*

*Reviewer #3: (No Response)*

*2. Is the manuscript technically sound, and do the data support the conclusions?*

*The manuscript must describe a technically sound piece of scientific research with data that supports the conclusions. Experiments must have been conducted rigorously, with appropriate controls, replication, and sample sizes. The conclusions must be drawn appropriately based on the data presented. *

*Reviewer #1: Yes*

*Reviewer #2: Yes*

*Reviewer #3: No*

*3. Has the statistical analysis been performed appropriately and rigorously? *

*Reviewer #1: Yes*

*Reviewer #2: Yes*

*Reviewer #3: No*

*4. Have the authors made all data underlying the findings in their manuscript fully available?*

*The PLOS Data policy requires authors to make all data underlying the findings described in their manuscript fully available without restriction, with rare exception (please refer to the Data Availability Statement in the manuscript PDF file). The data should be provided as part of the manuscript or its supporting information, or deposited to a public repository. For example, in addition to summary statistics, the data points behind means, medians and variance measures should be available. If there are restrictions on publicly sharing data—e.g. participant privacy or use of data from a third party—those must be specified.*

*Reviewer #1: Yes*

*Reviewer #2: Yes*

*Reviewer #3: No*

*5. Is the manuscript presented in an intelligible fashion and written in standard English?*

*PLOS ONE does not copyedit accepted manuscripts, so the language in submitted articles must be clear, correct, and unambiguous. Any typographical or grammatical errors should be corrected at revision, so please note any specific errors here.*

*Reviewer #1: Yes*

*Reviewer #2: Yes*

*Reviewer #3: Yes*

*6. Review Comments to the Author*

*Please use the space provided to explain your answers to the questions above. You may also include additional comments for the author, including concerns about dual publication, research ethics, or publication ethics. (Please upload your review as an attachment if it exceeds 20,000 characters)*

*Reviewer #1: Implementing this method in resource poor setting will be desirable, kindly take action for the same. Best wishes.*

*Reviewer #2: (No Response)*

*Reviewer #3: The authors were unable to address the comments. As the previously advised to include the clinical details of subjects enrolled in the study but in revised manuscript the data is missing. Moreover the cohort 1 and 2 are not clearly defined. Inclusion and exclusion criteria not mentioned for subject recruitment. Comparison of test with culture finding not calculated. Health control other than patients not included in the study as this is crucial part to evaluate/validate any test for disease detection.*

*7. PLOS authors have the option to publish the peer review history of their article (what does this mean?). If published, this will include your full peer review and any attached files.*

**

*Reviewer #1: No*

*Reviewer #2: No*

*Reviewer #3: **Yes: **Amit Singh*

**

*While revising your submission, please upload your figure files to the Preflight Analysis and Conversion Engine (PACE) digital diagnostic tool, https://pacev2.apexcovantage.com/. PACE helps ensure that figures meet PLOS requirements. To use PACE, you must first register as a user. Registration is free. Then, login and navigate to the UPLOAD tab, where you will find detailed instructions on how to use the tool. If you encounter any issues or have any questions when using PACE, please email PLOS at figures@plos.org. Please note that Supporting Information files do not need this step.*

---

## [Author Response · Author response to Decision Letter 1]

8 Jul 2024

We appreciate reviewer 3's careful review of our manuscript and have crafted the following response to each of reviewer 3's comments. This response is also found in the "Response to Reviewers" attachment. 

Reviewer comment 1: “As the previously advised to include the clinical details of subjects enrolled in the study but in revised manuscript the data is missing.”

Author response: We appreciate the reviewer’s reminder to include the baseline characteristics of the participants in our study. Since the initial submission of our manuscript, our group has published a related manuscript by Wood et al. [1] which evaluated swabs from the same group of participants. This published manuscript includes a table (Table 1) that describes the baseline characteristics of the participants from cohorts 1 and 2 including age, gender, race, HIV, status, and other relevant clinical details. 

In light of this, we have added a sentence to the revised manuscript (lines 221–223) indicating that the clinical details of the subjects enrolled in this study can be found in the manuscript by Wood et al. [1]. We believe this provides the necessary information and maintains consistency between our publications. 

Reviewer comment 2: “Moreover the cohort 1 and 2 are not clearly defined. Inclusion and exclusion criteria not mentioned for subject recruitment.”

Author response: We appreciate the reviewer’s concern regarding the definition of cohorts 1 and 2. While we have previously defined these cohorts in our manuscript (lines 217–221), we understand the need for more detailed descriptions. The Wood et al. [1] study provides comprehensive definitions of cohorts 1 and 2, including detailed inclusion and exclusion criteria. We added a reference to this publication in our manuscript for those who require further information. Specifically, we have added the following sentence to the methods section (lines 221–223): "Detailed definitions of Cohorts 1 and 2, including participant baseline characteristics and inclusion and exclusion criteria, are described in Wood et al."

Reviewer comment 3: “Comparison of test with culture finding not calculated.”

Author response: Thank you for your comment regarding the use of sputum GeneXpert Ultra to determine TB-positive samples in our study.

While sputum culture is traditionally regarded as the gold standard for TB diagnosis, the GeneXpert Ultra assay is endorsed by the World Health Organization (WHO) [2] and is considered a valid method for bacteriologically confirming TB cases. As stated by the WHO: “People diagnosed with TB using rapid molecular tests recommended by the WHO, lateral flow urine lipoarabinomannan (LF-LAM) assays, sputum smear microscopy or culture are defined as 'bacteriologically confirmed' cases of TB” [3] .

However, we acknowledge the importance of including culture data for comprehensive validation. To address this concern, we will revise our manuscript to include a statement acknowledging this limitation. Specifically, we have added the following sentence to the discussion of limitations in lines 413–414: “Our study is also limited in that culture data was not collected for Xpert Ultra-positive participants.” We hope this clarification and the addition of the limitation statement to our manuscript address your concerns.

Reviewer comment 4: “Health control other than patients not included in the study as this is crucial part to evaluate/validate any test for disease detection.”

Author response: We understand the concern regarding the selection of control subjects in our study. However, we would like to highlight a couple points that justify our methodology. 

Our choice to include symptomatic but TB-negative individuals in our control group was made deliberately to avoid inaccurately reporting our specificity due to spectrum bias. Spectrum bias is the misrepresentation of a test’s performance due to a study including too narrow a range of individuals with or without the disease of interest [4]. A review of meta-analyses of diagnostic tests [5] found that including severe cases and healthy controls resulted in a relative diagnostic odds ratio (RDOR) of 4.9, indicating a considerable overestimation of diagnostic accuracy due to the extreme contrast between groups. Using fully healthy controls (i.e., no TB-like symptoms) would likely lead to fewer false positive results, thus overestimating the specificity. 

Our inclusion of symptomatic, TB-negative subjects as controls aligns with the recommendations from the STARD guidelines [6] and PRISMA which state that primary studies of diagnostic test accuracy should recruit individuals that represent the population in which the test will be typically used. The primary purpose of our test is to distinguish between TB-positive and TB-negative individuals in a clinical setting where TB is suspected, which includes patients presenting with TB-like symptoms. Including truly healthy controls (individuals without any symptoms) would not accurately reflect the clinical scenarios in which the diagnostic test is applied.

In conclusion, we believe that the inclusion of symptomatic TB-negative controls is a methodologically sound approach that enhances the relevance and applicability of our findings. We hope this clarification addresses your concerns. 

Works Cited:

1. Wood RC, Luabeya AK, Dragovich RB, Olson AM, Lochner KA, Weigel KM, et al. Diagnostic accuracy of tongue swab testing on two automated tuberculosis diagnostic platforms, Cepheid Xpert MTB/RIF Ultra and Molbio Truenat MTB Ultima. Journal of Clinical Microbiology. 2024 Mar 14;62(4):e00019-24. 

2. Organization WH. WHO consolidated guidelines on tuberculosis: module 3: diagnosis: rapid diagnostics for tuberculosis detection [Internet]. World Health Organization; 2021 [cited 2024 Jul 1]. Available from: https://iris.who.int/handle/10665/342331

3. 2.2 Diagnostic testing for TB [Internet]. [cited 2024 Jul 1]. Available from: https://www.who.int/teams/global-tuberculosis-programme/tb-reports/global-tuberculosis-report-2023/tb-diagnosis---treatment/2.2-diagnostic-testing-for-tb

4. Usher-Smith JA, Sharp, SJ, Griffin SJ. The spectrum effect in tests for risk prediction, screening, and diagnosis. BMJ. 2016 Jun 22;353:i3139. 

5. Rutjes AWS, Reitsma JB, Di Nisio M, Smidt N, van Rijn JC, Bossuyt PMM. Evidence of bias and variation in diagnostic accuracy studies. CMAJ. 2006 Feb 14;174(4):469–76. 

6. Bossuyt PM, Reitsma JB, Bruns DE, Gatsonis CA, Glasziou PP, Irwig L, et al. STARD 2015: an updated list of essential items for reporting diagnostic accuracy studies. BMJ. 2015 Oct 28;351:h5527.

---

## [Decision Letter · Decision Letter 2]

19 Jul 2024

Detection of *Mycobacterium tuberculosis* from tongue swabs using sonication and sequence-specific hybridization capture**

*PONE-D-23-25589R2*

*Dear Dr. Yan,*

*We’re pleased to inform you that your manuscript has been judged scientifically suitable for publication and will be formally accepted for publication once it meets all outstanding technical requirements.*

*Within one week, you’ll receive an e-mail detailing the required amendments. When these have been addressed, you’ll receive a formal acceptance letter and your manuscript will be scheduled for publication.*

*An invoice will be generated when your article is formally accepted. Please note, if your institution has a publishing partnership with PLOS and your article meets the relevant criteria, all or part of your publication costs will be covered. Please make sure your user information is up-to-date by logging into Editorial Manager at Editorial Manager® and clicking the ‘Update My Information' link at the top of the page. If you have any questions relating to publication charges, please contact our Author Billing department directly at authorbilling@plos.org.*

*If your institution or institutions have a press office, please notify them about your upcoming paper to help maximize its impact. If they’ll be preparing press materials, please inform our press team as soon as possible -- no later than 48 hours after receiving the formal acceptance. Your manuscript will remain under strict press embargo until 2 pm Eastern Time on the date of publication. For more information, please contact onepress@plos.org.*

*Kind regards,*

*Atul Vashist, PhD*

Academic Editor

*PLOS ONE*

* *

*Additional Editor Comments (optional):*

* *

*Reviewers' comments:*

*Reviewer's Responses to Questions*

*

**Comments to the Author**
*

*1. If the authors have adequately addressed your comments raised in a previous round of review and you feel that this manuscript is now acceptable for publication, you may indicate that here to bypass the “Comments to the Author” section, enter your conflict of interest statement in the “Confidential to Editor” section, and submit your "Accept" recommendation.*

*Reviewer #3: All comments have been addressed*

*2. Is the manuscript technically sound, and do the data support the conclusions?*

*The manuscript must describe a technically sound piece of scientific research with data that supports the conclusions. Experiments must have been conducted rigorously, with appropriate controls, replication, and sample sizes. The conclusions must be drawn appropriately based on the data presented. *

*Reviewer #3: Yes*

*3. Has the statistical analysis been performed appropriately and rigorously? *

*Reviewer #3: N/A*

*4. Have the authors made all data underlying the findings in their manuscript fully available?*

*The PLOS Data policy requires authors to make all data underlying the findings described in their manuscript fully available without restriction, with rare exception (please refer to the Data Availability Statement in the manuscript PDF file). The data should be provided as part of the manuscript or its supporting information, or deposited to a public repository. For example, in addition to summary statistics, the data points behind means, medians and variance measures should be available. If there are restrictions on publicly sharing data—e.g. participant privacy or use of data from a third party—those must be specified.*

*Reviewer #3: Yes*

*5. Is the manuscript presented in an intelligible fashion and written in standard English?*

*PLOS ONE does not copyedit accepted manuscripts, so the language in submitted articles must be clear, correct, and unambiguous. Any typographical or grammatical errors should be corrected at revision, so please note any specific errors here.*

*Reviewer #3: Yes*

*6. Review Comments to the Author*

*Please use the space provided to explain your answers to the questions above. You may also include additional comments for the author, including concerns about dual publication, research ethics, or publication ethics. (Please upload your review as an attachment if it exceeds 20,000 characters)*

*Reviewer #3: The revised manuscript submitted by the authors addressed all the comments raised by me. The comments which could not be able to address are mentioned in limitations of the study.*

*7. PLOS authors have the option to publish the peer review history of their article (what does this mean?). If published, this will include your full peer review and any attached files.*

**

*Reviewer #3: **Yes: **Dr. Amit Singh*

---

## [Editor Report · Acceptance letter]

7 Aug 2024

PONE-D-23-25589R2 

PLOS ONE

Dear Dr. Yan, 

I'm pleased to inform you that your manuscript has been deemed suitable for publication in PLOS ONE. Congratulations! Your manuscript is now being handed over to our production team.

Kind regards, 

on behalf of

Dr. Atul Vashist 

Academic Editor

PLOS ONE